# Molecular epidemiology of *Mycobacterium tuberculosis* complex in the Volta Region of Ghana

**Selassie Ameke**[1,2], **Prince Asare**[1], **Samuel Yaw Aboagye**[1], **Isaac Darko Otchere**[1], **Stephen Osei-Wusu**[1], **Dorothy Yeboah-Manu**[1,2], **Adwoa Asante-Poku**[1,2] *

**1** Bacteriology Department, Noguchi Memorial Institute for Medical Research (NMIMR), University of Ghana, Accra, Ghana, **2** West African Centre for Cell Biology of Infectious Pathogens (WACCBIP), University of Ghana, Accra, Ghana

* aasante-poku@noguchi.ug.edu.gh

**Data Availability Statement:** Data cannot be shared publicly because of patient confidentiality. Data are available from the Noguchi Memorial Institutional Data Access / Ethics Committee

## Abstract

### Context

Available molecular epidemiological data from recent studies suggest significant genetic variation between the different lineages of *Mycobacterium tuberculosis* complex (MTBC) and the MTBC lineages might have adapted to different human populations.

### Aim

This study sought to determine the population structure of clinical MTBC isolates from the Volta Region of Ghana.

### Methods

The MTBC isolates obtained from collected sputum samples were identified by PCR detecting of *IS6110* and genotyped using spoligotyping. Non-tuberculous mycobacterial isolates were characterized by amplification of the heat shock protein 65 (*hsp65*) gene and sequencing. The drug susceptibility profiles of the MTBCs determined using GenoType MTBDRplus.

### Results

One hundred and seventeen (117, 93.6%) out of 125 mycobacterial positive isolates were characterized as members of the MTBC of which *M. tuberculosis* sensu stricto (MTBss) and *M. africanum* (MAF) were respectively 94 (80.3%) and 23 (19.7%). In all, 39 distinct spoligotype patterns were obtained; 26 for MTBss and 13 for MAF lineages. Spoligotyping identified 89 (76%) Lineage 4, 16 (13.6%) Lineage 5, 7 (6.0%) Lineage 6, 3 (2.6%) Lineage 2, 1(0.9%) Lineage 3 and 1 (0.9%) Lineage 1. Among the Lineage 4 isolates, 62/89 (69.7%) belonged to Cameroon sub-lineage, 13 (14.7%) Ghana, 8 (9.0%) Haarlem, 2 (2.2%) LAM, 1 (1.1%) Uganda I, 1 (1.1%) X and the remaining two (2.2%) were orphan. Significant localization of MAF was found within the Ho municipality (n = 13, 29.5%) compared to the more cosmopolitan Ketu-South/Aflao (n = 3, 8.3%) (p-value = 0.017). Eight (8) non-tuberculous mycobacteria were characterized as *M. abscessus* (7) and *M. fortuitum* (1).

(contact via GWemakor@noguchi.ug.edu.gh) for researchers who meet the criteria for access to confidential data.

**Funding:** This work was supported by WACCBIP Postdoctoral Fellowship funds to A.A.P from a DELTAS Africa grant (DEL-15-007: Awandare). The DELTAS Africa Initiative is an independent funding scheme of the African Academy of Sciences (AAS) Alliance for Accelerating Excellence in Science in Africa (AESA) and supported by the New Partnership for Africa's Development Planning and Coordinating Agency (NEPAD Agency) with funding from the Wellcome Trust (107755/Z/15/Z: Awandare) and the UK government. The views expressed in this publication are those of the author(s) and not necessarily those of AAS, NEPAD Agency, Wellcome Trust or the UK government. SM was supported by Wellcome Trust intermediate fellowship grant 097134/Z/11/Z to Dorothy Yeboah-Manu.

**Competing interests:** The authors have declared that no competing interests exist.

## Conclusion

We confirmed the importance of *M. africanum* lineages as a cause of TB in the Volta region of Ghana.

## Introduction

Tuberculosis (TB) still remains an important global public health problem and continues to pose great burden on the healthcare systems of many developing countries especially in Sub-Saharan Africa [1]. Due to the worldwide emergence of multidrug-resistant TB strains and the increasing burden of HIV, TB is gradually becoming untreatable. In 2019, an estimated 10 million people contracted TB with 1.3 million TB-related deaths placing TB among the top ten causes of death worldwide [1]. In Ghana, TB still poses a public health challenge with a TB incidence rate of 148/100,000 population per year, is ranked the 19[th] most TB-burdened country in Africa by WHO [1]. In 2017, Ghana together with Angola, Democratic Republic of Congo, Ethiopia, Kenya, Uganda and South Africa, constituted high TB-HIV burden countries in Africa [1].

*Mycobacterium tuberculosis sensu stricto* (MTBss) and *Mycobacterium africanum* (MAF) are the major pathogenic species of the *M. tuberculosis* complex (MTBC) in humans [2, 3]. There are 8 lineages of the MTBC, Lineages 1 to 4,7 and 8 belonging to MTBss whereas L5 and L6 belonging to MAF. Improved genomic analysis disproves previous dogma of genomic homogeneity of these lineages but indicates that there are significant variation with functional implications. We now also know that these lineages exhibit a phylogeographical structure with specific lineages being associated with distinct geographical areas [4] suggesting potential host-pathogen interaction. This could influence the broad applicability of control tools such as diagnostics and vaccine [4].

*M. africanum* (MAF) is endemic only in West Africa causing up to 40% of TB in some West African settings. MAF is considered less virulent compared to other human TB causing pathogens [5, 6] thus, expected to be outcompeted by the more virulent MTBss over time [5]. However, findings from recent studies still indicate significant presence of MAF within West Africa [6–8]. One reason for the high prevalence of MAF might be due to the stable adaptation of this lineage to some human populations. Two independent molecular epidemiological studies from our group, found a strong association between MAF and an indigenous West African ethnic group (Ewe ethnicity) which was driven by MAF L5 [9, 10]. A follow up comparative genomics studies, found MAF lineages (L5 and L6) to be completely different pathogens [2]. The genome of MAF L5 indicated a pathogen of limited host range compared to L6, which gave an indication of wide host range [11]. The Volta region is the traditional home of the Ewe ethnicity in Ghana; however, relative to the other regions of the country, to the best of our knowledge no study on the population structure of prevailing strains have been done [8, 12]. We characterized MTBC isolates obtained from patients attending specific health facility in the Volta region of Ghana to determine the circulating genotypes and drug resistance.

## Subjects and methods

### Study design, ethical clearance and case recruitment

This was a one-year cross-sectional study. All protocols used for this study were reviewed and approved by the Institutional Review Board of the Noguchi Memorial Institute for

Medical Research with federal assurance number FWA00001824. Written informed consent was sought from all participants but for minors under 18 years, consent was sought from their parents or legal guardians. The objectives of the study and procedures were explained carefully to all study participants before inclusion into the study. The procedure for sputum sample collection for routine diagnosis of TB in Ghana was followed. A structured questionnaire was used to obtain standard demographic and epidemiologic data of patients. Two hundred and seventy (270) consented newly diagnosed smear-positive pulmonary TB patients agreed to be included in the study. Collected samples were stored at 4˚C and transported within 4 days to Noguchi Memorial Institute for Medical Research (NMIMR) for laboratory analysis.

## Study area and patients' characteristics

The study was conducted in 12 public health facilities located in the Volta Region of Ghana which together reports more than 90% of all TB cases in Volta Region as indicated in **Fig 1.** Approximately 70% of the inhabitants of the study area are of the Ewe ethnicity [13] with main occupation being crop and fish farmers along the Volta Lake.

## Isolation of mycobacterial species from sputa

Sputum samples were decontaminated using the 5% oxalic acid decontamination method and inoculated on 4 Lowenstein–Jensen (L-J) media (2 L-J media supplemented with glycerol and 2 with pyruvate) and incubated at 37˚C until macroscopic growth was observed as previously described [14]. Direct smear microscopy was performed for confirmation of acid-fast bacilli (AFB) using Ziehl-Neelsen (ZN) staining.

## Isolation of genomic DNA

A loop full of mycobacterial colonies confirmed as AFB growing at the log phase was suspended in 1mL of sterile distilled water and inactivated by heating at 95˚C for 1 hour to disrupt mycobacterial cell wall to release DNA into suspension. The resulting suspension was stored at -20˚C and used for all downstream DNA-based assays.

## Genotyping of MTBC

Mycobacteria isolates were confirmed as members of the MTBC by PCR amplification of the MTBC-specific insertion sequence 6110 (*IS6110).* All *IS6110* positive samples were genotyped by spoligotyping as previously described using primers DRa (5′-CCG AGA GGG GAC GGA AAC-3′) and biotinylated DRb (5′-GGT TTT GGG TCT GAC GAC-3′) [15]. In Brief, the amplified DNA was tested for the presence of specific spacers by hybridization with a set of 43 oligonucleotides derived from the spacer sequences of *M. tuberculosis* H37Rv and *M. bovis* BCG P3 (the GenBank accession no. for the sequence of *M. tuberculosis* H37Rv is Z48304, and that for *M. bovis* BCG P3 is X57835). Bound fragments were revealed by chemiluminescence after incubation with horseradish peroxidase-labeled streptavidin (Boehringer Mannheim). Negative water controls were PCR amplified and included on each blot to identify any possible amplicon contamination. In addition, positive controls (H37Rv and *M. bovis* BCG DNA) was amplified and included on each blot. Shared types were defined as patterns common to at least two or more isolates. All patterns that could not be assigned were considered orphan spoligotypes. Spoligotypes were analysed as character types. The obtained spoligotyping patterns were compared with those available in the international spoligotype database SITVIT and SpolDB4 databases containing 35,925 spoligotypes comprising 39,295 isolates from 122 countries. A

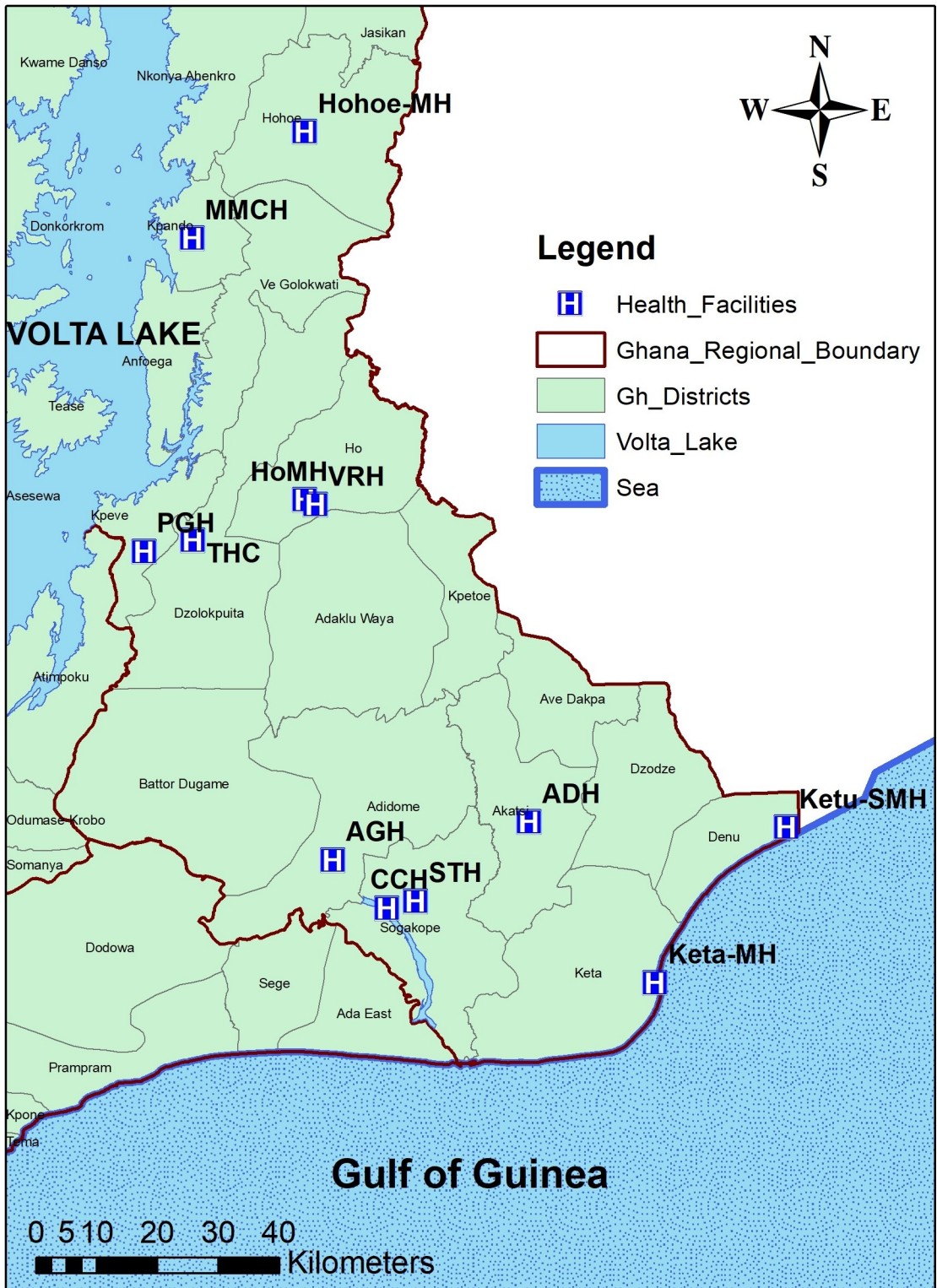

**Fig 1. Study location and 12 health facilities selected.** ADH: Akatsi District Hospital, AGH: Adidome Government Hospital, CCH: Comboni Catholic Hospital, HoMH: Ho Municipal Hospital, Hohoe-MH: Hohoe Municipal Hospital, Keta-MH: Keta Municipal Hospital, Ketu-SMH: Ketu South Municipal Hospital, MMCH: Margret Marquardt Catholic Hospital, VRH: Volta Regional Hospital, STH: South Tongu Hospital, THC: Tsito Health Center, PGH: Peki Government Hospital.

shared type was defined as a spoligotyping pattern common to at least two isolates, and clades were assigned according to signatures described in the database. Relationships among the isolates were inferred from Spoligotyping using the both SITVIT and SpolDB4 databases. All IS*6110*-negative isolates were further characterized by PCR amplification and sequencing of the mycobacterial specific heat shock protein (*hsp*) 65 with the primers TB11: 5'-ACC AAC GAT GGT GTG TCC AT-3' and TB12: 5'-CTT GTC GAA CCG CAT ACC CT- 3' as previously described [15, 16].

## Drug susceptibility testing by line probe assay

Confirmed MTBC isolates were screened for their susceptibility to isoniazid (INH) and rifampicin (RIF) using the GenoType MTBDR*plus* version 2.0 (Hain Lifesciences) according to the manufacturer's protocol (Hain Lifesciences, 2015). Drug resistance was expressed as the absence of wild-type band, presence of mutation band or both.

## Data analysis

Information from the structured questionnaire was entered into Microsoft excel and validated. Statistical analyses such as Chi-square and fisher's exact test were carried out using STATA SE 12 with p-values of less than 0.05 at 95% confidence considered significant. Spoligotype patterns were entered in a binary format in Microsoft excel, uploaded and compared with those available in the online database http://www.pasteur-guadeloupe.fr:8081/SITVIT_ONLINE and http://www.miru-vntrplus.org useful in molecular typing of *Mycobacterium tuberculosis* complex. The genetic relationships among all of the identified circulating spoligotype patterns were studied by constructing a dendrogram using the categorical parameter and the UPGMA coefficient available in the *miru-vntrplus* online tool (SpolDB4).

# Results

## Study population

From January 2016 to January 2017, 270 participants were enrolled into the study. One hundred and eighty-three were confirmed as acid-fast bacilli positive (67.8%), of which 125 isolates were obtained from culture (68%). Of the 125 isolates obtained, 8 (6.4%) were identified as non-tuberculous mycobacteria whereas 2 (1.6%) could not be typed after repeated analysis and were thus excluded from further analysis. Hence, a total of 115 MTBC isolates were used for the downstream analysis. As indicated in **Table 1**, age of patients ranged from 12 to 86 years with a mean age of 44.8 years ± 14.9 years. Thirty-five (30.4%) of the cases were females and the remaining 80 (70.1%) being males. Majority of the participants 107 (93.0%) were mainly of the Ewe ethnicity. The main occupation were, traders 44/115 (38.31%) and farmers 35/115 (30.4%). All 115 TB patients consented to HIV testing, and sero-positivity was 3.5%.

## Population structure of the MTBC isolates

Based on spoligotyping, we identified six out of the seven human-associated MTBC lineages in our study population (**Table 2**). Lineage distributions are 89 (76%) Lineage 4, 16 (13.6%) Lineage 5, 7 (6.0%) Lineage 6, 3 (2.6%) Lineage 2, 1 (0.9%) Lineage 3 and 1 (0.9%) Lineage 1. Among the Lineage 4 isolates, 62/89 (69.7%) belonged to Cameroon sub-lineage, 13 (14.7%) Ghana, 8 (9.0%) Haarlem, 2 (2.2%) LAM, 1 (1.1%) Uganda I, 1 (1.1%) X and the remaining two (2.2%) were orphan. Among the 89 L4, 62/89 (69.7%) belonged to the Cameroon sub-lineage (mostly the spoligotype with shared international type (SIT) number 61). In addition to the Cameroon family, four other sub-lineages namely 13 (14.7%) Ghana,

**Table 1. Characteristics of MTBC positive participants.**

| Gender (115) | Value (percentage) |
|---|---|
| Male | 80 (69.6) |
| Female | 35 (30.4) |
| **Age (115)** | |
| Mean ± SD | 44.8 ± 14.9 years |
| Median (IQR) | 43 (35–54) years |
| **Occupation (115)** | |
| Traders | 42 (36.5) |
| Farmer | 34 (29.6) |
| Artisans | 9 (7.8) |
| Drivers | 5 (4.3) |
| Students | 4 (3.5) |
| Teacher | 2 (1.7) |
| Hospital Orderlies | 1 (0.9) |
| Unemployed | 15 (13.0) |
| **Religion (115)** | |
| Christianity | 107 (93.0) |
| Islam | 4 (3.5) |
| Traditional | 2 (1.7) |
| No Religion | 2 (1.7) |
| **Ethnicity (115)** | |
| Ewe | 107 (93.0) |
| Guan | 3 (2.6) |
| Hausa | 3 (2.6) |
| Akan | 1 (0.9) |
| Fulani | 1 (0.9) |
| **Nationality (115)** | |
| Ghanaians | 112 (97.4) |
| Togolese | 2 (1.7) |
| Malian | 1 (0.9) |
| **HIV Status (115)** | |
| Negative | 93 (80.9) |
| Positive | 4 (3.5) |
| Not Done | 18 (15.6) |
| **Previously Treated (115)** | |
| No | 111 (96.5) |
| Yes | 4 (3.5) |

8 (9.0%) Haarlem, 1 (1.1%) Uganda I, 2 (2.2%) LAM, 1 (1.1%) X and the remaining two (2.2%) were orphan (**Table 2**). Overall, we identified 39 distinct spoligotyping patterns among the 117 MTBC isolates analysed. Twenty-one unique patterns (singletons) and 18 clustered patterns comprising of 96 isolates were also identified. The odds of an isolate belonging to a cluster were higher among MTBss compared to MAF (OR = 4.39 CI = 1.56–12.35). Cameroon sub-lineage of MTBss strain gave the largest cluster with 44 isolates sharing a spoligotype (SIT 61). In addition, we identified 23 novel spoligotypes among our isolates compared to the SITVIT database (**Fig 2**).

**Table 2. Prevalence of *Mycobacterium tuberculosis* complex lineages and sub-lineages.**

| Species (N = 115) | Lineages | Sub-Lineages | Number (%) |
|---|---|---|---|
| MTBss (91, 79.1%) | Lineage 2 | Beijing | 3 (2.6%) |
| | Lineage 3 | Delhi/CAS | 1 (0.9%) |
| | Lineage 4 (87, 75.7%) | Cameroon | 64 (55.6%) |
| | | Ghana | 11 (9.6%) |
| | | Haarlem | 9 (7.8%) |
| | | LAM | 2 (1.7%) |
| | | X | 1 (0.9%) |
| MAF (24, 20.9%) | Lineage 5 | West Africa I | 17 (14.8%) |
| | Lineage 6 | West Africa II | 7 (6.1%) |

## Spatial distribution of MTBC lineages and sub-lineages among MTBC Isolates

The combined number of isolates analysed from the different geographical areas, together with identified species, lineages and sub-lineages are indicated in Fig 3. As expected MTBss dominated in all sites, and was the only lineage identified in Hohoe(n = 14), Sogakofe(n = 3) and Kpando (n = 1). We found significant difference in inter-municipality comparisons. For instance, driven by Lineage 5, the proportions of MAF in Ho Municipality, 13/44 (29.6%) showed a significantly higher proportion (p-value = 0.017) than in Ketu-South Municipality (Aflao), 3/36 (8.3%) **(Fig 3)**.

## Prevalence of drug resistance among the MTBC isolates

A total of 101 MTBC isolates were analyzed by GenoType® MTBDR*plus* version 2.0 following the manufacturer's instructions. We found 6 (5.9%), 5 (4.9%) and 2 (1.9%) of the isolates to be INH mono-resistant, RIF mono-resistant and multidrug resistant (MDR) respectively

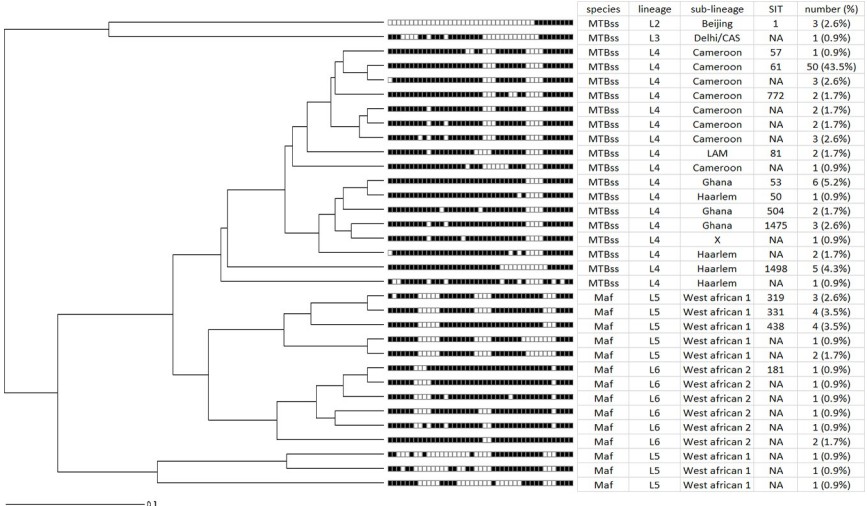

**Fig 2. Relationship of identified 33 spoligotype profiles.** Tree was plotted using the MIRU-VNTR*plus* web application available at https://www.miru-vntrplus.org/.

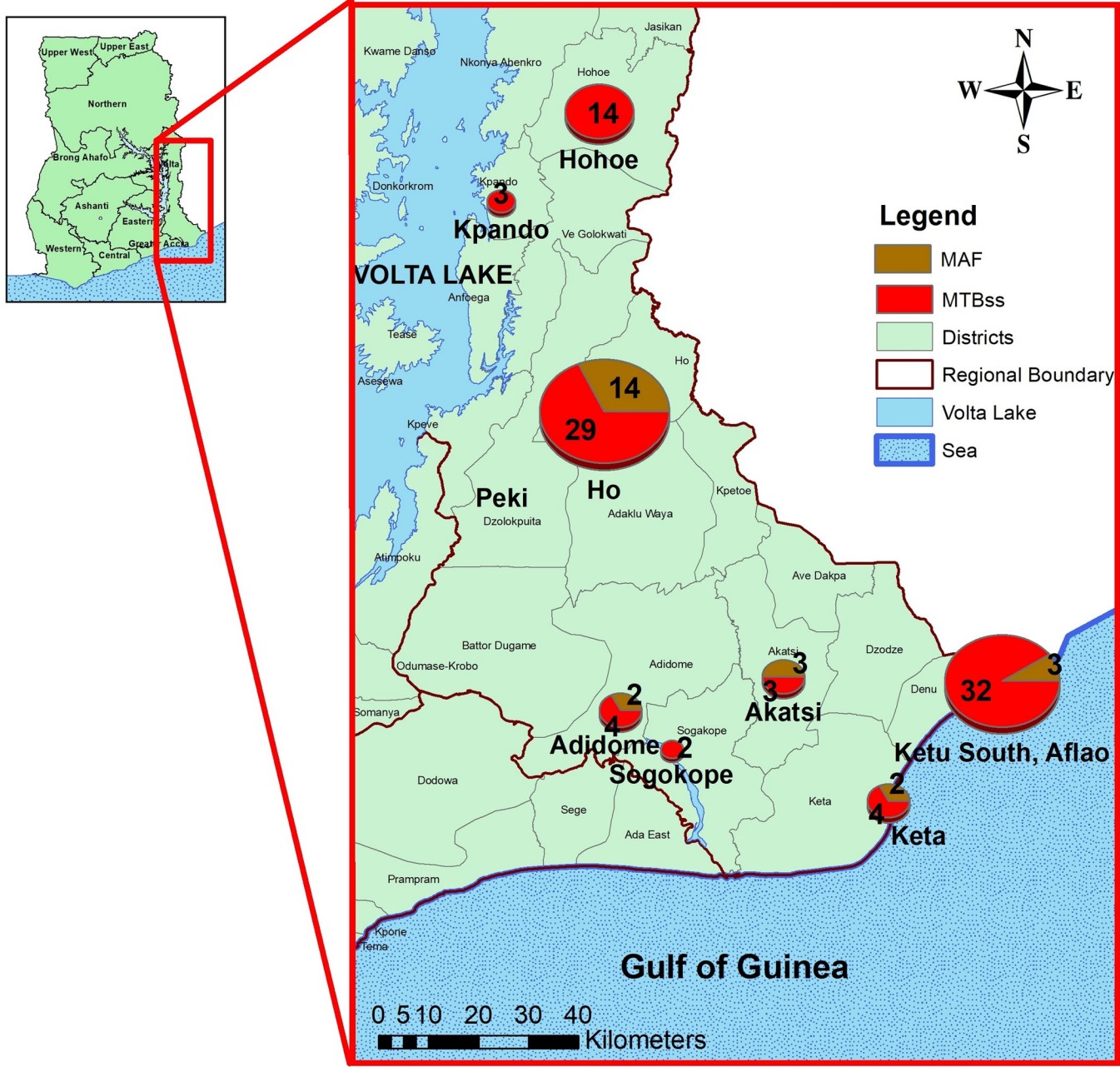

**Fig 3. Geospatial distribution of the two human adapted MTBC species; *M. tuberculosis* sensu stricto (MTBss) and *M. africanum* (MAF).** Figure was generated in ArcGIS. **Permissions**: The authors permits unrestricted use, distribution, and reproduction in any medium, provided the original author and source are credited.

(**Table 3**) Stratifying our dataset by genotypes, five (6.41%) of the MTBss isolates were found to be INH mono-resistant while only one (4.55%) MAF isolate was resistant to INH. All the RIF-mono resistant and the two MDR isolates were found to be MTBss. All two MDR isolates showed the MUT3 band for *rpoB* gene which corresponds to the SNP C1592U that resulted in the locus amino acid change, S531L. One of the MDR isolates in addition to the above had the MUT2B band which is associated with the locus amino acid change, H526D. None of the RIF-

**Table 3. Drug susceptibility profile of the human adapted MTBC isolates amongst TB patients.**

| Species | INH | | | | RIF | | | | MDR |
|---|---|---|---|---|---|---|---|---|---|
| | Sensitive | Resistant | OR (CI) | p-value | Sensitive | Resistant | OR (CI) | p-value | |
| MTBss (N = 75) | 67 (89.3%) | 8 (10.7%) | 2.75 (0.33–127.06) | 0.3350 | 68 (90.7%) | 7 (9.3%) | 2.37 (028–111.27) | 0.4189 | 3 (4.0%) |
| MAF (N = 24) | 23 (95.8%) | 1 (4.2%) | | | 23 (95.8%) | 1 (4.2%) | | | 0 (0%) |
| Total (N = 99 | 93 (93.9%) | 6 (6.1%) | | | 94 (94.9%) | 5 (5.1%) | | | 3 (3.0%) |

mono resistant isolates showed a mutation band but rather absence of wild-type bands. Majority of the RIF-mono resistant isolates had the *rpoB* wild-type band, WT1 absent, and its absence is associated with any of these loci amino acid changes, F505L, T508A or S509T. Similarly, an absence of *rpoB* wild-type band WT8 in three isolates corresponds with any of these loci amino acid changes, S531L, S531W or L533P. The isoniazid resistance among the MDRs was conferred by *KatG* mutant MUT1 that corresponds to the SNP U943A that resulted in the S315T locus amino acid change. The *KatG* mutation dominated in 4 MTBss isolates and was responsible for majority of INH resistance. However, *inhA* mutants MUT1 and MUT3B associated with the locus amino acid changes C15T, and T8A on the promoter region were also implicated in INH resistance as shown in **Table 3**.

## Identified nontuberculous mycobacteria species

The 8 AFB positive isolates that were IS*6110* negative were identified by *hsp65* gene sequencing followed by NCBI Blast search. Seven of them as *M. abscessus* and the remaining one as *M. fortuitum*.

## Discussion

We sought to determine the population structure of MTBC isolates obtained from smear positive pulmonary patients attending public health facilities in the Volta Region. We found unique spatial distribution of 6 lineages of MTBC with MAF being responsible for 19.7% of cases, odds of (INH)-mono resistance, (RIF)-mono resistance and MDR higher for MTBss infection and identification of NTMs in 8 TB cases.

Previous studies in Ghana reported the prevalence of 6 human-adapted MTBC lineages in the Greater Accra, Central and the Northern regions of Ghana [10, 14]. In this study we found these 6 MTBC lineages also circulating in the Volta Region suggesting their establishment in Ghana. Comparing the proportion of MAF to the national prevalence of 20% we observed no significant difference (p-value = 0.951) indicating that the same proportion is circulating in the Volta Region [14]. Approximately 20% MAF proportion has been found in previous studies to be fairly stable over an 8-year period [8].

*M. africanum* is an important cause of human TB in West Africa, causing about 50% of all TB cases reported in some West African countries. Past epidemiological surveys saw a dramatic drop of MAF numbers in several West African countries [17, 18], suggesting a replacement by the more virulent MTBss. However in sharp contrast, two recent studies from several regions in Nigeria (Abuja, Ibadan, Nnewi and Cross River State) estimated persistently high MAF prevalence between 14% and 33% and detected foci of recent transmission [19–21].

One possible reason for the stability of MAF in Ghana and West Africa irrespective of the observed lower virulence might be adaptation of this lineage to specific human populations. Recently, two independent molecular epidemiological studies conducted in Ghana found a strong association of MAF with the Ewe ethnicity [9, 10]. Since Volta region is the home of the Ewe ethnic group, we decided to assess the distribution of MAF within this region. Although we

expected to find a greater proportion of MAF in the Volta region, more so when most of the participants were of Ewe ethnicity, interestingly we found significant difference in inter-municipality comparisons. For instance, driven by L5, the proportions of MAF in the Ewe dominated Ho Municipality, 13/44 (29.6%) showed a significantly higher proportion than in Ketu-South Municipality (Aflao), 3/36 (8.3%) (p-value = 0.017). The significantly lower prevalence of MAF in the Ketu-South Municipality may be due to the diverse human populations as a result of travelers and migrants present at every point in time crossing the border to and from the Republic of Togo. This observation of significant association of MAF with the Ewes could be an indication of a possible predisposing factor among this human population to MAF infection.

Drug resistance remains a great threat to the fight against TB. Using proportional method, Asante-Poku *et al*., (2015) (10) and Homolka et al., (2010) [12] recorded high INH mono resistance. Other studies have reported high level INH resistance (40–95%) to be associated with 75–90% katG position S315T mutation [22, 23]. Using Line probe assay, our study observed 40% high level INH resistance that was associated with only 57.14% katG position S315T mutation. Our finding was consistent with findings by Otchere et al., (2016) [11], which showed that the human adapted strain MTBss compared to MAF has a relatively greater risk of possessing this position S315T mutation in katG (p < 0.001). Riccardi et al., (2009) [24], associated RIF resistance with the rpoB gene mutations which cluster mainly in the codon region of 507–533. Although inconsistent with the mutation in rpoB S450L reported by Otchere et al. (2016) [11], our findings showed rpoB gene mutation distribution of 37.5% S531L and 12.5% H526D. The amino acid change from polar serine to non-polar leucine at position 531 and from basic histidine to aspartic acid at position 526 may have contributed to conformational change in protein structure. This may have subsequently prevented proper binding of the drug RIF to the β-subunit of the DNA dependent RNA polymerase leading to drug resistance.

Differential diagnosis of MTBCs and Non-tuberculous mycobacteria (NTMs) is very crucial for the appropriate treatment regimen to be administered [25–27]. While the standard treatment regimen for MTBC infection takes 6 months, NTMs therapy however takes between 18–24 months with different drug regimen based on thorough drug susceptibility testing as the NTMs are naturally resistant to majority of anti-TB drugs [25]. In this study, NTMs were isolated from 6.4% of patients presumptively diagnosed with TB using the NTP diagnostic algorithm. This was higher than 2.5% observed by Otchere et al. [28], but consistent with studies by Bertoletti et al., (2011) [29]. The NTMs isolated, *M. abscessus*, and *M. fortuitum* are known to be fast growing mycobacteria that can cause pulmonary infections in both immunocompetent and immunocompromised individuals. Microscopy, which is used for TB diagnosis in the periphery medical laboratories, lacks specificity and is unable to distinguish between MTBCs and NTMs. These observations support the need to pay critical attention to differential diagnosis of pulmonary infectious mycobacteria, most especially in the rolling out of DNA-based diagnostics especially among cases that do not sputum convert after two months of anti-TB treatment to allow appropriate management of such cases.

In conclusion, our study confirms the importance of MAF in Ghana and highlight the need to incorporate MAF studies into development of TB control tools.

## Acknowledgments

We express our gratitude to all laboratory staff and study participants of the various health facilities for their time and cooperation during the study period.

## Author Contributions

**Conceptualization:** Dorothy Yeboah-Manu, Adwoa Asante-Poku.

**Data curation:** Dorothy Yeboah-Manu, Adwoa Asante-Poku.

**Formal analysis:** Selassie Ameke, Prince Asare, Samuel Yaw Aboagye, Isaac Darko Otchere, Stephen Osei-Wusu, Dorothy Yeboah-Manu, Adwoa Asante-Poku.

**Funding acquisition:** Dorothy Yeboah-Manu, Adwoa Asante-Poku.

**Investigation:** Selassie Ameke, Prince Asare, Dorothy Yeboah-Manu, Adwoa Asante-Poku.

**Methodology:** Selassie Ameke, Dorothy Yeboah-Manu, Adwoa Asante-Poku.

**Project administration:** Dorothy Yeboah-Manu, Adwoa Asante-Poku.

**Resources:** Dorothy Yeboah-Manu.

**Writing – original draft:** Selassie Ameke, Dorothy Yeboah-Manu, Adwoa Asante-Poku.

**Writing – review & editing:** Selassie Ameke, Prince Asare, Samuel Yaw Aboagye, Isaac Darko Otchere, Stephen Osei-Wusu, Dorothy Yeboah-Manu, Adwoa Asante-Poku.

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
