## [Decision Letter · Decision Letter 0]

13 Nov 2020

PONE-D-20-26015

Molecular Epidemiology of Mycobacterium tuberculosis complex in Volta Region of Ghana

PLOS ONE

Dear Dr. Aasante-Poku,

Thank you for submitting your manuscript to PLOS ONE. After careful consideration, we feel that it has merit but does not fully meet PLOS ONE’s publication criteria as it currently stands. Therefore, we invite you to submit a revised version of the manuscript that addresses the points raised during the review process.

A major revision is required of all parts of the manuscript to improve presentation, data analysis and interpretation as recommended by the reviewers. In particular, do use the most recent SITVIT2 and not outdated SpolDb4.

We look forward to receiving your revised manuscript.

Kind regards,

Igor Mokrousov, Ph.D., D.Sc.

Academic Editor

PLOS ONE

Journal Requirements:

2. Thank you for your title change from "phylogenetic..." to "molecular epidemiology..."

Please also revise the text of your manuscript to remove references to phylogenetics in the following places:

a) Key Message

b) Last paragraph of your Introduction

c) End of your Genotyping of MTBC section

d) Figure 2 legend

4. Please include a copy of Table 5 which you refer to in your text on page 12.

Reviewers' comments:

Reviewer's Responses to Questions

**Comments to the Author**

1. Is the manuscript technically sound, and do the data support the conclusions?

Reviewer #1: Partly

Reviewer #2: Yes

2. Has the statistical analysis been performed appropriately and rigorously? 

Reviewer #1: Yes

Reviewer #2: Yes

3. Have the authors made all data underlying the findings in their manuscript fully available?

Reviewer #1: Yes

Reviewer #2: Yes

4. Is the manuscript presented in an intelligible fashion and written in standard English?

Reviewer #1: No

Reviewer #2: Yes

5. Review Comments to the Author

Reviewer #1: The authors present important and timely findings which will add to the gaps in knowledge on the molecular epidemiology of TB in Africa and Ghana to be precise. The manuscript however, will require revising as recommended below:

-Ensure that formatting and grammar is checked throughout the entire document, there are several grammatical errors throughout the document, i.e. lines 37 and 44, no full stops after a sentence, line 73 two comas after Ethiopia, line 78 check spacing before and after 7.

-Abstract: Instead of the statement "standard methods" list the methods used during analysis refer to line 42. Decimal places should be consistent throughout the manuscript i.e. line 49 is presented as 76.04% instead of 76.06% which should be rounded to 76.1% to maintain one decimal place throughout the manuscript. Check that the calculations for percentages are correct, percentages are not adding up to 100%, refer to lines 49 and 50. List percentages for the two orphans. It will be useful to add Ghana and spoligotyping to the list of key words and either limit to tuberculosis and MAF in order to maintain the journal limit.

-Some statements will need to be rephrased in order for them to be more cohesive i.e. lines 68/69 rephrase the sentence beginning "The worldwide.." to "Due to the worldwide". Insert "TB"after multidrug-resistant in line 68. Rephrase line 71 to "...public health challenge; with a TB incidence rate of...". Check line 72 "a is ranked...". Line 74 insert "and" between Uganda and South Africa, delete "the" before Africa in line 74.

-Study design: line 136 should be changed from "on patients" to "of patients". Line 137, state how many samples were included, were all samples included? and how samples were selected for analysis. line 150, there shouldn't be a space between 37 and degree C. DRb and not "Drb"line 165. Spacing between bovis and BCG line 167. lines 171 to 174 "in order... each blot" are not necessary, authors can delete the statement and simply refer to the reference. Line 175 should make reference to the database use to assign SIT and orphans. line 178 refers to the SpolDB4, however the SITVIT2 is a more updated version of the SpolDB4 which the authors should use to compare their spoligo findings and make reference to. Authors should cite manufacturers correctly line 188.

-Data analysis: the titles data analysis and results (lines 192 and 201) are underlined which is not consistent with the remainder of the manuscript. Was SITVIT2 or SpolDB not used for analysis? line 197.

-Results: The sample size should also be mentioned earlier, somewhere in the study design i.e. line 137. Line 204, only 125 out of 270 (46%) positive isolates were characterised, why is this the case? This impacts on the conclusions drawn as it is not a good representation of isolates from this region, should be mentioned later as a limitation to the study. Delete "The" from the title in line 216. Was phenotypic DST data available? If so this should be included in table 1. Lines 218-219, lineage statistics don't add up to 100% and are not matching with the data presented in the abstract. Line 244 table number not indicated. Line 252 "wide type band" shouldn't be in italics.

-Discussion: line 278, refrain from listing i), ii)... Line 287 should read "Approximately..." and not "The approximately...". A full stop after ref (14) in line 287. Line 289, it is not advisable to start a sentence with an abbreviation "MAF". In your discussion and conclusion you will need to address the fact that only 46% of isolates collected over the study period were characterised and how this would impact on the data that you have presented. Line 310, delete "pulmonary" since drug resistant TB can impact treatment outcome EPTB as well as PTB. et al should be in italics throughout including in reference list. Abbreviate M. abscessus in line 333 unless this was the first use. Rephrase the statement "The challenge is, microscopy..."in line 335. There is a big jump from speaking about diagnostics to concluding on MAF genotype, you will need to add a few sentences that connect the previous statement with your conclusion line 341.

-References: Check the reference style is uniform throughout your reference list and conforms with the journal requirements, italicize "et al". Delete () in line 373

-Figures: Is figure 1 the authors own work? If not cite the source. Figure 2 percentages are not adding up to 100%.

Reviewer #2: It was interesting reading your manuscript even similars studies have been done in the country in the same subject except for the Volta Region which add a value to the manuscript. I noticed minor corrections regarding the form for several lines: you need to adjust the space between some words see line 37;44;73;78;99;167:204;207;228;235;244;273;313;319

Please make M.africanum in italic in all the manuscript for example line 62.

Also a coma must be place in line 260 after table 3 and in line 287 after the refererence 14.

some references are old like the TB report which date is 2018, new TB report are available please use it

References content some errors which need to be correct: references 2;4;6;9;10;12;16;25;29;30;31

Table 3 must be redone to improve understanding because it is rather poorly presented

6. PLOS authors have the option to publish the peer review history of their article (what does this mean?). If published, this will include your full peer review and any attached files.

Reviewer #1: **Yes: **Namaunga Kasumu Chisompola

Reviewer #2: No

---

## [Author Response · Author response to Decision Letter 0]

23 Dec 2020

editor

All comments has been addressed

Reviewer 1

 all comments and edits have been addressed

Reviewer2

all comments and edits have been addressed

---

## [Editor Report · Decision Letter 1]

30 Dec 2020

PONE-D-20-26015R1

Molecular Epidemiology of Mycobacterium tuberculosis complex in Volta Region of Ghana

PLOS ONE

Dear Dr. Aasante-Poku,

Thank you for submitting your manuscript to PLOS ONE. After careful consideration, we feel that it has merit but does not fully meet PLOS ONE’s publication criteria as it currently stands. Therefore, we invite you to submit a revised version of the manuscript that addresses the points raised during the review process.

Before consider the following changes before I can proceed with your manuscript:

1.Your Cover letter is presented in the format with visible track changes mode. Please submit a **clean **version of cover letter.

2. Your answers to the editor and reviewers provided in this way are **not **acceptable. 

"editor

All comments has been addressed

Reviewer 1

all comments and edits have been addressed

Reviewer2

all comments and edits have been addressed"

**The answers should be detailed and point-to-point; it is also advised in each answer to refer to the particular page and line numbers in the revised version.**

We look forward to receiving your revised manuscript.

Kind regards,

Igor Mokrousov, Ph.D., D.Sc.

Academic Editor

PLOS ONE

---

## [Author Response · Author response to Decision Letter 1]

26 Feb 2021

Editor

1.Please ensure that that your manuscript meets PLOS ONE's style requirements, including those for file naming. 

 Response: The Manuscript has been revised to meet PLOS ONE style requirement 

2. Thank you for your title change from "phylogenetic..." to "molecular epidemiology..."

Please also revise the text of your manuscript to remove references to phylogenetics in the following places:

a) KeyMessage

b) Last paragraph of your Introduction

c) End of your Genotyping of MTBC section

d) Figure 2 legend 

 Response: The Manuscript text has been revised to remove references to “phylogenetic in the key areas suggested 

3.We note that you have indicated that data from this study are available upon request. PLOS only allows data to be available upon request if there are legal or ethical restrictions on sharing data publicly. For information on unacceptable data access restrictions, please see http://journals.plos.org/plosone/s/data-availability#loc-unacceptable-data-access-restrictions.

We will update your Data Availability statement on your behalf to reflect the inform

 Response:The cover letter has been revised to reflect the ethical restrictions on the data set. 

4. Please include a copy of Table 5 which you refer to in your text on page 12. 

 Response: Table 5 has been removed from the manuscript accordingly 

Reviewers comment 

Reveiwer 1

1.Ensure that formatting and grammar is checked throughout the entire document, there are several grammatical errors throughout the document, i.e. lines 37 and 44, no full stops after a sentence, line 73 two commas after Ethiopia, Line 78 check spacing before and after 7 

 Response: We thank the reviewer for the discovery of these errors.The grammatical errors throughout the document corrected accordingly 

2. Abstract: Instead of the statement "standard methods" list the methods used during analysis refer to line 42. 

 Response:Standard methods haven clarified. Line 41-42

3. Decimal places should be consistent throughout the manuscript i.e. line 49 is presented as 76.04% instead of 76.06% which should be rounded to 76.1% to maintain one decimal place throughout the manuscript. 

 Response: Decimal places corrected throughout the manuscript to one decimal place. Line 49

4. Check that the calculations for percentages are correct, percentages are not adding up to 100%, refer to lines 49 and 50. 

 Response: All percentages corrected throughout the manuscript to one decimal place. Line 52

5. List percentages for the two orphans. 

 Response: Percentages of the two orphans listed. Line 52

6. It will be useful to add Ghana and spoligotyping to the list of key words and either limit to tuberculosis and MAF in order to maintain the journal limit. 

 Response: Ghana and spoligotyping added to the key words Line 58

7. Some statements will need to be rephrased in order for them to be more cohesive i.e.

 lines 68/69 rephrase the sentence beginning "The worldwide." to "Due to the worldwide". 

 Response: The worldwide corrected Due to the worldwide" Line 70

8. Insert "TB"after multidrug-resistant in line 68. 

 Response: TB" inserted after multidrug-resistant. Line 70

9. Rephrase line 71 to "...public health challenge; with a TB incidence rate of...".

 Response : Line 71 corrected to read: "with a TB incidence rate of". Line 73

10. Check line 72 "a is ranked...".

 Response: "a" deleted Line 74

11. Line 74 insert "and" between Uganda and South Africa, delete "the" before Africa in line 74 

 Response: "and" inserted between Uganda and South Africa. Line 76

12. Study design: line 136 should be changed from "on patients" to "of patients". Line 137.

 Response: line 136 changed from "on patients" to "of patient. Line 142

13. State how many samples were included, were all samples included? and how samples were selected for analysis. 

 Response: All samples collected were included in the study. Line 142

14. line 150, there shouldn't be a space between 37 and degree C. 

 Response: Space between 37 and degree C corrected. Line 155

15. DRb and not "Drb"line 165. 

 Response: "Drb" corrected to DrB. Line 174

16. Spacing between bovis and BCG line 167. 

 Response: Spacing between bovis and BCG corrected. Line 176

17. lines 171 to 174 "in order... each blot" are not necessary, authors can delete the statement and simply refer to the reference. 

 Response: Text in Lines 171 to 174 deleted

18. Line 175 should make reference to the database use to assign SIT and orphans. line 178 refers to the SpolDB4, however the SITVIT2 is a more updated version of the SpolDB4 which the authors should use to compare their spoligo findings and make reference to. Authors should cite manufacturers correctly line 188.

 Response: Obtained spoligotyping patterns were compared with those available in the international spoligotype database SITVIT and SpolDB4 . Line 212

19. Data analysis: the titles data analysis and results (lines 192 and 201) are underlined which is not consistent with the remainder of the manuscript. 

 Response: We have duly undone the underlines. Line 207, 233

20. Was SITVIT2 or SpolDB not used for analysis? line 197.

 Response: Both SITVIT2 and SpolDB were used for analysis. Lime 212 

21. Results: The sample size should also be mentioned earlier, somewhere in the study design i.e. line 137. 

 Response: sample size indicated in the study design. Line 142

22. Line 204, only 125 out of 270 (46%) positive isolates were characterised, why is this the case? This impacts on the conclusions drawn as it is not a good representation of isolates from this region, should be mentioned later as a limitation to the study. 

 Response: Thank you very much for the comment. However, 270 participants were enrolled into the study. One hundred and eighty-three were confirmed as acid-fast bacilli positive (67.8%), of which 125 culture positives (68%) where obtained. The results has been corrected to reflect this statement.. Line 235-239

23. Delete "The" from the title in line 216. 

 Response: "The deleted from the Title. Line 249

24. Was phenotypic DST data available? If so this should be included in table 1. 

 Response: DST was performed for all isolates obtained. The results has been presented in a Table 3. Line 300

25. Lines 218-219, lineage statistics don't add up to 100% and are not matching with the data presented in the abstract. 

 Response: Percentages corrected to match the data presented in the abstract. Line 258

26. Line 244 table number not indicated.

 Response:Table number indicated. Line 258

27. Line 252 "wide type band" shouldn't be in italics.

 Response: "wild-type band" has been un- italised. Line 289 

28. Discussion: line 278, refrain from listing i), ii)... 

 Response:Numbering of key results has been deleted. Line 311-314

30. Line 287 should read "Approximately..." and not "The approximately...". 

 Response: Text corrected accordingly. Line 325

31. A full stop after ref (14) in line 287. 

 Response: Full stop added accordingly. Line 320

32. Line 289, it is not advisable to start a sentence with an abbreviation "MAF". 

 Response:MAF written out in full. Line 293

33. In your discussion and conclusion you will need to address the fact that only 46% of isolates collected over the study period were characterised and how this would impact on the data that you have presented. 

 Response:Thank you very much for the comment. However, 270 participants were enrolled into the study. One hundred and eighty-three were confirmed as acid-fast bacilli positive (67.8%), of which 125 culture positives (68%) where obtained. The results has been corrected to reflect this statement. Line 248-250

34. Line 310, delete "pulmonary" since drug resistant TB can impact treatment outcome EPTB as well as PTB. 

 Response:As suggested by reviewer "pulmonary" deleted from the sentence/ Line 410

35. et al should be in italics throughout including in reference list. 

 Response: et al italised throughout the reference list

36. Abbreviate M. abscessus in line 333 unless this was the first use.

 Response:Mycobacterium abscessus abbreviated to M. abscessus in text. Line 377

37. Rephrase the statement "The challenge is, microscopy..."in line 335.

 Response:The sentence has been rephrased. Line 377

38. There is a big jump from speaking about diagnostics to concluding on MAF genotype, you will need to add a few sentences that connect the previous statement with your conclusion line 341.

 Response: Thank you for your comments. A few Sentences has been added 

39. References: Check the reference style is uniform throughout your reference list and conforms with the journal requirements, italicize "et al". 

 Response: Reference style corrected. Line 515 

40.Delete () in line 373

 Response: () deleted. Line 450 

41. Figures: Is figure 1 the authors own work? If not cite the source. 

 Response:Thank you for the comment. Figure 1 is the authors own work 

42.Figure 2 percentages are not adding up to 100%.

 Response: Percentages corrected for Figure 2.

Reviewer 2

1.you need to adjust the space between some words see line 37;44;73;78;99;167:204;207;228;235;244;273;313;319

 Response: Topographic errors have been corrected through out the manuscript

2. Please make M.africanum in italic in all the manuscript for example line 62.

 Response:M.africanum has been italised through the manuscript. Line 62

3.Also a coma must be place in line 260 after table 3 and in line 287 after the refererence 14

 Response:Thank you for the comment. A full stop has been placed in Line 266 after Table 3 and Line 287. Line 295, 320

4.some references are old like the TB report which date is 2018, new TB report are available please use it

 Response: Thank you for the comment. Referece changed to reflect new TB report

5.References content some errors which need to be correct: references 2;4;6;9;10;12;16;25;29;30;31

 Response:Reference content corrected

6.Table 3 must be redone to improve understanding because it is rather poorly presented

 Response:Thank you for the comment. Table 3 has been redone. Line 300

---

## [Editor Report · Decision Letter 2]

1 Mar 2021

Molecular Epidemiology of Mycobacterium tuberculosis complex in Volta Region of Ghana

PONE-D-20-26015R2

Dear Dr. Aasante-Poku,

We’re pleased to inform you that your manuscript has been judged scientifically suitable for publication and will be formally accepted for publication once it meets all outstanding technical requirements.

Kind regards,

Igor Mokrousov, Ph.D., D.Sc.

Academic Editor

PLOS ONE
---

## [Editor Report · Acceptance letter]

9 Mar 2021

PONE-D-20-26015R2 

**Molecular epidemiology of *Mycobacterium tuberculosis* complex in the Volta Region of Ghana**

Dear Dr. Aasante-Poku:

I'm pleased to inform you that your manuscript has been deemed suitable for publication in PLOS ONE. Congratulations! Your manuscript is now with our production department. 

Kind regards, 

on behalf of

Dr Igor Mokrousov 

Academic Editor

PLOS ONE